# Hybrid Signal-Processing Method Based on Neural Network for Prediction of NO_3_, K, Ca, and Mg Ions in Hydroponic Solutions Using an Array of Ion-Selective Electrodes

**DOI:** 10.3390/s19245508

**Published:** 2019-12-13

**Authors:** Woo-Jae Cho, Hak-Jin Kim, Dae-Hyun Jung, Hee-Jo Han, Young-Yeol Cho

**Affiliations:** 1Department of Biosystems and Biomaterial Engineering, College of Agriculture and Life Sciences, Seoul National University, Seoul 08826, Korea; er8er@snu.ac.kr (W.-J.C.); jeoguss@gmail.com (D.-H.J.); honeyjo@snu.ac.kr (H.-J.H.); 2Research Institute of Agricultural and Life Sciences, College of Agriculture and Life Sciences, Seoul National University, Seoul 08826, Korea; 3Smart Farm Research Center, Korea Institute of Science and Technology (KIST), Gangneung-si 25451, Gangwon-do, Korea; 4Major of Horticultural Science, College of Applied Life Sciences, Jeju National University, Jeju 63243, Korea; yycho@jejunu.ac.kr

**Keywords:** ion-selective electrode, sensor array, two-point normalization, artificial neural network, hydroponics

## Abstract

In closed hydroponics, fast and continuous measurement of individual nutrient concentrations is necessary to improve water- and nutrient-use efficiencies and crop production. Ion-selective electrodes (ISEs) could be one of the most attractive tools for hydroponic applications. However, signal drifts over time and interferences from other ions present in hydroponic solutions make it difficult to use the ISEs in hydroponic solutions. In this study, hybrid signal processing combining a two-point normalization (TPN) method for the effective compensation of the drifts and a back propagation artificial neural network (ANN) algorithm for the interpretation of the interferences was developed. In addition, the ANN-based approach for the prediction of Mg concentration which had no feasible ISE was conducted by interpreting the signals from a sensor array consisting of electrical conductivity (EC) and ion-selective electrodes (NO_3_, K, and Ca). From the application test using 8 samples from real greenhouses, the hybrid method based on a combination of the TPN and ANN methods showed relatively low root mean square errors of 47.2, 13.2, and 18.9 mg∙L^−1^ with coefficients of variation (CVs) below 10% for NO_3_, K, and Ca, respectively, compared to those obtained by separate use of the two methods. Furthermore, the Mg prediction results with a root mean square error (RMSE) of 14.6 mg∙L^−1^ over the range of 10–60 mg∙L^−1^ showed potential as an approximate diagnostic tool to measure Mg in hydroponic solutions. These results demonstrate that the hybrid method can improve the accuracy and feasibility of ISEs in hydroponic applications.

## 1. Introduction

Hydroponics is a cultivation method that grows plants using nutrient solutions composed of water and nutrient salts without soil. Recently, hydroponics has been widely and rapidly utilized in agricultural industries because it is the most intensive and effective production method that can be designed to support year-round production with high yields and good quality [1,2]. Hydroponics is usually classified into open and closed types. In open hydroponics, the nutrient solution flows through the growing bed and is discarded, which can result in the pollution of ground- and surface water [3,4]. In closed hydroponics, which collects drainage solutions and reuses these by replenishing water and nutrients, the use and discharge of water and nutrients are less than for open hydroponics [5,6,7]. Therefore, a transition from open hydroponics to closed hydroponics is seen increasingly often due to the more environmentally-friendly aspect of closed hydroponics [7]. However, current practices for closed hydroponics still have several limitations, as described below.

In closed hydroponics, the management of the reused solutions is mostly conducted by the conductivity and pH probes. However, the probes can only provide a total ion activity and pH, so the imbalance of nutrient ratios may occur in reused nutrient solutions due to the lack of information about the individual ion concentrations [8,9,10,11,12]. This makes the crop quality and productivity decrease. Therefore, growers usually flush the nutrient solutions and replace all solutions periodically, despite the environmental pollution and loss of fertilizers [13]. Although growers can analyze the individual ion concentrations of the nutrient solutions by periodic laboratory analysis, a time delay between the sampling and the analysis limits the instantaneous feedback control of the nutrient solution composition [14,15]. In this regard, fast and continuous measurement of individual nutrient concentrations is necessary for the precise correction of the reused nutrient solutions, thereby allowing both improved efficiency of fertilizer use and reduced environmental pollution [9,16,17].

Ion-selective electrodes (ISEs) could be one of the most attractive tools to measure the individual ion concentrations of hydroponic solutions due to their advantages such as simplicity of use, fast response time, direct measurement of analyte, sensitivity over a wide concentration range, and portability [18,19,20]. Specifically, the concept of a sensor array makes it possible to simultaneously determine individual ion concentrations in complex samples [9,21,22]. However, several disadvantages of ISEs such as signal drift and distortions due to interfering ions make the application for hydroponics difficult [9,13,17,23]. Therefore, it is essential to develop an effective data-processing method to compensate for the signal drift and interference [24,25].

One such method is a two-point normalization (TPN) method in conjunction with the use of the Nernst equation that consists of a sensitivity adjustment followed by an offset adjustment applied to all of the signal data measured with the ISEs [18,23,26]. In previous studies, the TPN method was employed and shown to be effective in compensating for the signal drifts of a sensor array consisting of NO_3_, K, and Ca ISEs which were used for measuring hydroponic solutions [10,17,18,23,26,27]. However, the TPN method is relatively weak for the interference because the standard curve for the TPN is constructed based on the simplified Nernst equation. The simplified Nernst equation assumes the ion-selective membrane would be specific to the ion of interest, but the actual membrane responds to other interfering ions. As a result, electromotive forces (EMFs) generated from the ISEs are affected by the complex ion matrix in hydroponic solutions, thereby inducing errors in the ion concentrations predicted by the TPN method. In addition, use of the TPN method is still limited in measuring other ions, such as P and Mg, present in hydroponic solutions, because ionophores for selective recognition of the P and Mg with an acceptable level are not yet commercially available.

Considering the complexity of ions present in hydroponic solutions, an artificial neural network (ANN) would be a proper method for compensating for the interferences on ISEs because ANN conducts the processing of non-linear multivariate interactions based on knowledge storage and learning and its property of controlling the number of hidden neurons and hidden layers makes it more flexible than other machine-learning techniques [15,25,28,29,30]. In addition, ANN could be utilized as a predictive tool through the reflection of inherent chemical relationships [28]. However, ANN is vulnerable to signal drifts. For example, drifts can make the signals different from the signals obtained during the training, then the predicted ion concentrations by the ANN model would deviate from the actual values. This indicates that the ANN model would be difficult to use in ISE measurements without the drift compensation [17].

Based on the complementary properties of the TPN and ANN methods, in this study, we proposed a hybrid signal processing approach to effectively compensate for the signal drifts and interferences from other ions, thereby improving the accuracy of ISEs in hydroponic applications. The specific objectives of this study were (1) to evaluate the hybrid processing method when compared to the TPN or ANN methods using a sensor array consisting of ion-selective electrodes for macronutrients (NO_3_, K, and Ca) and an electrical conductivity electrode, and (2) to investigate the possibility of an ANN-based prediction model for Mg concentration in hydroponic solutions, of which there are few robust ISEs.

## 2. Materials and Methods

### 2.1. Preparation of the Sensor Array

For the measurement of NO_3_ and K ions, two different polyvinyl chloride (PVC)-based ion-selective membranes were formulated based on the chemical compositions previously reported (Table 1) [18,23,26]. The ion-selective membrane solutions were prepared by dissolving the chemicals with 2 mL of tetrahydrofuran (THF) solvent. The solutions were then poured into a 24-mm diameter glass ring (48953, Sigma-Aldrich, St. Louis, MO, USA) with a flat glass plate (48952, Sigma-Aldrich, St. Louis, MO, USA) and evaporated for 24 h at room temperature. When the solutions were evaporated, ion-selective membrane films were punched with a diameter of 2.5 mm. The punched films were attached to the ends of laboratory-made plastic bodies of 44 mm length using THF solvent. As a final step, the internal solutions, consisting of 0.01 M NaNO_3_ + 0.01 M NaCl for NO_3_ ISEs, and 0.01 M KCl for K ISEs, were filled. For sensing Ca ions, a commercially available Ca ISE (Orion 9320BNWP, Thermo Fisher Scientific, Beverly, MA, USA) was used. A double junction glass electrode (Orion 900200, Thermo Fisher Scientific, Beverly, MA, USA) was used as the reference electrode for ISEs. In addition, a commercial conductivity probe (Orion 013610MD, Thermo Fisher Scientific, Beverly, MA, USA) was employed to measure the conductivity of the test samples. Finally, the sensor array was composed of three ISEs for NO_3_, three ISEs for K, two ISEs for Ca, one reference electrode, and one conductivity probe. It has been reported that the ISEs prepared in the study are applicable for hydroponic solutions [10,17,18,23,26,27]. The performance characteristics of the ISEs reported in the previous studies are summarized in Table 2.

### 2.2. Construction and Evaluation of Data-Processing Methods

Two conventional processing methods (TPN and ANN) were used and compared to validate the feasibility of the hybrid processing method (TPN-ANN). The working principle of the TPN method is that individual sensitivity slopes of each of the ISE electrodes are normalized by multiplying the EMF data by the ratio of a reference EMF difference to a measured EMF difference using two different solutions with known concentrations of the primary ion corresponding to the electrodes. Offsets are then adjusted by subtraction of the difference between the highest reference point and the modified highest concentration point. The EMF data modified by use of the TPN method are applied to the simplified Nernst equation (Equation (1)).
(1)$$EMF=E_{O}+E_{J}+S\log a_{i}$$
where *E_O_*, *E_J_*, *S*, and $a_{i}$ are the standard potential (mV), the liquid-junction potential (mV), Nernstian slope (59.16/*z_i_* mV/decade change in concentration for H_2_O at 25 °C and *z_i_* is the charge number of the response ion *i*), and the activity of the response ion.

The parameters of calibration equations determined in the previous study [23], i.e., *S*, *E_O_*, and *E_J_*, could be utilized because the compositions of ISE membranes were the same. According to the procedures in previous studies [17,18,23,26,27], the TPN was carried out prior to each sample measurement.

The structure of the ANN used in this study was a feed-forward backpropagation neural network, which consisted of an input layer, hidden layers, and an output layer. The numbers of neurons in the input layer and the output layer were 9 (signals from eight ISEs and one conductivity probe) and 4 (NO_3_, K, Ca, and Mg), respectively. Although ANNs with multiple hidden layers and neurons have a stronger generalization ability, the training time is usually increased and more samples are required to avoid an over-fitting issue [31]. Therefore, for the application of the ANN, the parameters of ANN such as the number of hidden layers or hidden neurons should be determined carefully.

The optimal numbers of hidden layers and neurons were determined via trial and error method. Briefly, the number of input neurons was fixed as 10 and the number of hidden layers was set to 1, 2, 3, 5, and 10. Three replicate results were then obtained for each layer number and their root mean square errors (RMSEs) were calculated and compared to select the optimal number of hidden layers. Similarly, the number of hidden neurons was tested using ranges of 8 to 16 with an interval of 2 because the neuron number is highly related to the predictability of the ANN model [31]. The model performance was evaluated based on RMSEs of three replicate training results.

During the learning process, the learning rate of 0.01 and the Levenberg–Marquardt algorithm, which is one of the optimizer algorithms for avoiding local minima and overfitting, were used [32]. The input data (X_s_) for ANN was rescaled (X_r_) using min-max scaling (Equation (2)) to make each input have equal meanings and dimensions for the neural network.
X_r_ = (X_s_ − X_min_)/(X_max_ − X_min_),
(2)
where X_min_ and X_max_ are the minimum value and the maximum value of the input dataset, respectively.

As a next step, a conversion of input values to output values was carried out to calculate the interconnections between input values and output values, which is called an activation function. Due to the non-linear interactions among the ISEs, non-linear activation functions such as the tanh (*tansig*) [33] and rectified linear unit (ReLU) [34,35] were considered for the hidden layer. Specifically, the application of the *tansig* showed the high accuracy in ISE signal processing in the previous study [9]. However, the *tansig* function limits the output range as −1 to 1. As a result, the output would be diminished when the hidden layer number is increased, thereby reducing the predictability of the ANN model. This problem is called the “vanishing information problem” [36]. ReLU makes the output sparser so it can be effective in multi-layer neural networks [32,33]. Therefore, ReLU was used for the neural networks of the hidden layers of 5 and 10.

After the determination of the parameters for the ANN, the original ANN was trained using the raw EMFs from the sensor array. In the case of the hybrid method, the ANN was applied using the EMFs after the TPN to achieve the drift compensation for the enhancement of the signal processing.

For the data processing, Python 3.7.3 programming language and several third-party libraries were used. The performances of the constructed processing methods were evaluated by the determination coefficients (R^2^) and RMSEs of the correlation between the predicted concentrations and the actual concentrations.

### 2.3. Preparation of Samples

Two-point normalization solutions and training samples were necessary to generate the primary information for the model training of TPN and ANN, respectively. Referring to the procedure described by the previous study [9], 27 solutions were designed by a fractional factorial design with three levels of concentration and four factors (NO_3_, K, Ca, and Mg) using a commercial statistical software (JMP, SAS Institute, Inc., Cary, NC, USA). Briefly, various mixtures of the primary ions (NO_3_, K, Ca, and Mg) were prepared to have concentrations of 100–1000, 30–300, 24–240, and 10–100 mg∙L^−1^ for NO_3_, K, Ca, and Mg, respectively, by adding the calculated stock solutions of ammonium nitrate, magnesium sulfate, potassium sulfate, and calcium chloride to a base solution. In order to generate training samples with a similar background of real hydroponic solutions, a mixture of the modified Hoagland’s hydroponic nutrient solution [37] and tap water (1/1 (*v*/*v*)) was used as the base solution for the training samples. The samples of the lowest levels and the highest levels of NO_3_, K, Ca, and Mg ions (i.e., 100 and 1000 mg L^−1^, 30 and 300 mg L^−1^, 24 and 240 mg L^−1^, 10 and 100 mg L^−1^, respectively) were additionally prepared for two-point normalization solutions.

For evaluating the feasibility of the processing methods in real hydroponic application, a total of 8 samples were manually collected from nutrient solution mixing tanks of various hydroponic systems (Table 3). Specifically, the samples had different compositions for six kinds of plants (kale, *Atractylodes japonica, Glehnia littoralis*, beet, basil, and paprika), which spanned a wide range of ion concentrations.

The actual concentrations of the samples were determined by a standard soil-water testing laboratory (National Instrumentation for Environmental Management (NICEM), Seoul, Korea) using an ion chromatograph (ICS-5000, Thermo Fisher Scientific, Waltham, MA, USA) with a low detection limit of 0.05 mg∙L^−1^ for NO_3_, and an inductively coupled plasma-optical emission spectrometer (iCAP 7400, Thermo Fisher Scientific, Waltham, MA, USA) with a detection limit of 0.6 μg∙L^−1^ for K, Ca, and Mg, respectively. The measured ion concentrations of the samples are shown in Table A1.

### 2.4. Procedure of Sample Measurements

In order to accurately obtain the signals from the sensor array and effectively apply the TPN prior to each sample measurement, a laboratory-made automated test stand modified from the system of the previous study was used [18,38]. The schematic diagram of the automated test stand is shown in Figure 1a. The test stand includes a Teflon-based sensor array chamber equipped with a servomotor, sample containers, a main computer system with a signal-conditioning data acquisition board, a motor controller, discrete pressure pumps for samples, and a control box for pump and motor operation (Figure 1b). The specifications of components in the test stand are listed in Table A2.

For each sample measurement, about 50 mL of sample solution was automatically injected into the sample holder by the pressure pumps and stirred by rotating the holder at approximately 30 rpm during data collection. Each test sequence began with a rinsing of the electrodes by introducing the distilled water (DW). Sixty seconds after the sample injection, the signals of the electrodes were logged with the mean of a 1 s burst of 1 kHz data. After each measurement, the holder was rinsed with distilled water and the rotational speed was increased to approximately 400 rpm to expel solutions centrifugally. The test sequence was controlled by software developed based on LabVIEW. Figure 2 represents the overall process of the sample measurements in this study. Three iterations were conducted for the prepared samples and Excel 2016’s statistical tools (Microsoft, Redmond, WA, USA) were used to analyze the data.

## 3. Results

### 3.1. Determination of the Artificial Neural Network (ANN) Structure

The RMSEs according to the hidden layers and the hidden neurons are shown in Figure 3. When the layer number was increased, the RMSEs of the prediction was increased (Figure 3a). Specifically, the ANN with single-hidden layer shows significantly low average RMSEs when compared to the ANN with multi-hidden layer. Therefore, the optimization of neuron numbers was conducted using a single hidden layer. In the same way, the number of neurons in the hidden layer was determined to be 14. The final structure of ANN used in this study is shown in Figure 4.

### 3.2. Evaluation of the Processing Methods in Training Samples

In the training step, the performances of the ANN-based processing methods (ANN and TPN-ANN) training and the TPN method were evaluated. The prediction results according to the processing methods are shown in Figure 5. In NO_3_ prediction (Figure 5a), the TPN showed a linear and accurate prediction result with R^2^ of 0.99, a slope of 0.87, and a RMSE of 89.1 mg∙L^−1^. In the case of the ANN and the hybrid method, there was no significant difference in the prediction results despite the lower RMSE of the hybrid method (ANN: 22.3 mg∙L^−1^, TPN-ANN: 19.2 mg∙L^−1^). Specifically, the highly linear relationships with R^2^ of 0.99 and slopes of 1.00 supported the proposition that the training of the ANN components would be well achieved.

The K prediction results showed similar trends in R^2^, slopes, and RMSEs (Figure 5b). The TPN method showed a good prediction result with an R^2^ of 0.99, a slope of 1.01, and an RMSE of 9.3 mg∙L^−1^. In the ANN training, the RMSE was 26.3 mg∙L^−1^, which was slightly higher than the RMSE of the TPN. However, the R^2^ of 0.97 and the slope of 0.94 showed the training was conducted at an acceptable level [29]. The TPN-ANN method showed improved training performance with a R^2^ of 0.99, a slope of 1.00, and an RMSE of 3.7 mg∙L^−1^.

In the Ca prediction results, it was remarkable that the ANN-based approaches had more stable and linear responses when compared to the TPN-based approach (Figure 5c). Specifically, the TPN showed a linear relationship with R^2^ of 0.82 and a slope of 1.57, a RMSE of 93.0 mg∙L^−1^, which was relatively high considering the Ca concentration of training samples ranging from 30 to 300 mg∙L^−1^. The ANN-based methods showed better performances with R^2^ of 0.97 and slopes of 0.97 and 0.96, and low RMSEs of 18.0 and 18.9 mg∙L^−1^ for the ANN and the TPN-ANN methods, respectively.

The Mg prediction result (Figure 5d) was only achieved by the ANN-based methods because the TPN has no predictability in ions without a directly related measurable sensor. The training results show that the ANN-based Mg prediction had a slope of 0.29, a R^2^ of 0.51, and a RMSE of 29.3 mg∙L^−1^. The result of the hybrid processing method showed an improved slope, R^2^, and RMSE, which were 0.4, 0.69, and 24.9 mg∙L^−1^, respectively. Although the values are somewhat subjective factors for evaluating the model performance, it would be possible to use the prediction model based on the hybrid method for the approximate quantitative prediction of Mg concentration according to the criteria of the previous study [39]. The correlation values between the predicted concentrations with the actual concentrations are presented in Table 4.

### 3.3. Application of the Processing Methods in Real Hydroponic Samples

After the training and evaluation of the processing methods in laboratory-made samples, the applicability of the processing methods for the sensor array was validated by the prediction of the ion concentrations of real hydroponic samples. Figure 6 shows the ion concentrations of the real hydroponic samples determined by the standard analyzers and the sensor array with the three processing methods. For NO_3_ and K concentrations, the ANN-based prediction was less accurate than the TPN-based prediction. Specifically, the ANN-based prediction made significant deviations (*p* < 0.01) in most sample measurements comparing the actual concentrations (Figure 6a,b). The hybrid method (TPN-ANN) predicted the concentration to be closer to the actual concentrations in NO_3_ and K than other methods, which indicated that the hybrid method improved the accuracy of the sensor array by effectively processing the signals.

When comparing the RMSEs obtained with the three methods (Table 5), even though the TPN showed lower RMSEs than those of the ANN (TPN: 75.4 and 19.8 mg∙L^−1^, ANN: 133.5 and 144.7 mg∙L^−1^ for NO_3_ and K, respectively), the hybrid method (TPN-ANN) showed the best predictability with RMSEs of 47.2 and 13.2 mg∙L^−1^_,_ and coefficients of variation (CVs) below 10% for NO_3_ and K, respectively. Moreover, in the Ca prediction (Figure 6 and Table 5), the RMSE of 18.9 mg∙L^−1^ obtained with the TPN-ANN was the lowest. In the Mg prediction, although the error bars showed relatively high CVs (26.6% and 28.6% for ANN and TPN-ANN methods), the Mg prediction results were almost comparable to the actual values, implying that the TPN-ANN method would offer the potential for use in hydroponic magnesium sensing.

Figure 7 shows changes in EMFs obtained with two-point normalization solutions (the high and low concentrations for NO_3_ (Figure 7a), K (Figure 7b), and Ca (Figure 7c), respectively) during the ANN training. As shown in the figures, the EMFs were varied over time, indicating the need for compensating for sensitivity and offset changes over time. In addition, the EMF differences obtained with the low and high concentration solutions, i.e., sensitivities, were nearly constant, implying that the use of the two-point normalization method would be effective in minimizing the signal drifts of all of the tested ISEs during the measurement. This confirmed a reason of worse predictabilities of the ANN compared to those obtained with the TPN and TPN-ANN methods might be related to no use of the TPN.

## 4. Discussion

In this study, we suggested a hybrid signal-processing method to improve the accuracy and feasibility of ISEs in hydroponic application by effectively compensating for the signal drifts and interferences from other ions.

The optimization result of the number of hidden layers showed a single hidden layer ANN had the lowest RMSE for NO_3_, K, Ca, and Mg prediction (Figure 3a). In actual fact, the ANN models with more hidden layers do not guarantee better performance than those with fewer layers if the number of hidden layers is sufficient for the given non-linear problem [40]. Similarly, the performance of the ANN model was not increased according to the number of hidden neurons (Figure 3b), as reported in the previous study [29].

In the training sample measurements (Figure 5, Table 4), the application of the TPN showed a strongly linear relationship with R^2^ of 0.99 despite a slight underestimation of NO_3_ concentrations between the actual and predicted concentrations similar to that reported in previous studies [23,26,27]. This might be caused by signal interferences from other ions, such as Cl and SO_4_ in training samples. Moreover, TPN-based Ca prediction has a deviated slope of 1.57, which might be due to the presence of Mg ions, which have a similar chemical behavior to Ca ions [41]. The cross-interference would affect the Nernstian slope, thereby inducing inaccuracy in prediction [42]. To solve the interference issue, the ANN, which would possibly compensate for the interfering responses by training the various backgrounds, was employed and improved the performance of the actual test (Figure 5a,c). It supports the theory that ANN would be effective for the non-linear interference by adjusting the relationship as reported in the previous studies [9,15,28,31,42]. In addition, we applied the ANN to predict the Mg concentration because we expected the ANN would extract the signals from the Mg ions through the training with defined background samples. Although the results were not satisfactory (Figure 5d, Table 4), the ANN-based models could be used to discriminate between high and low concentrations of Mg according to the criterion of the previous study [39].

In real sample application, the TPN-ANN was the best processing method, followed by TPN and ANN (Figure 6, Table 5). As mentioned above, the Ca prediction by the TPN was vulnerable to interferences. Although the TPN made Ca predictions more precise than the ANN in several samples, e.g., Basil 1, *Atractylodes japonica,*
*Glehnia littoralis* 1, and *Glehnia littoralis* 2, relatively high variations in Ca predictions depending on the samples showed that the TPN could be affected by the changes of background ions. In contrast, the ANN-based methods were effective in managing the interferences in actual tests, showing they were less affected by the samples in most cases (Figure 6c). However, the ANN method showed high RMSEs in the predictions of NO_3_, K, and Mg. The main reason for the errors in the ANN-based prediction would be due to the signal drifts. This limitation of the ANN was similar to the results of several studies [9,15,29]. The TPN method proved its effectiveness in drift compensation with improved accuracy. However, there were deviations in Ca prediction similar to those in the training sample measurement, which could be due to the interference by the various background ions. The hybrid method showed the best predictability in real hydroponic sample application by successfully combining the strengths of the TPN and the ANN, as expected. It meant the hybrid method could compensate for the signal drifts and then calculate the concentrations considering the non-linear influences from the interference through the neural network. As a result, the hybrid method improved the accuracy and the precision of the prediction of the ion concentrations with the lowest RMSEs of 47.2, 13.2, and 18.9 mg∙L^−1^ and CVs below 10% for NO_3_, K, and Ca, respectively.

In Mg prediction, the RMSE of 29.4 mg∙L^−1^ in the ANN-based prediction is high considering the range of 10–60 mg∙L^−1^ in real samples. However, by applying the hybrid method, the RMSE of the prediction was reduced to 14.6 mg∙L^−1^. Considering the lack of the ISEs for the direct measurement of Mg, it would be possible to improve the predictability by adding more ISEs which are more closely related to the Mg ion.

## 5. Conclusions

In this study, a hybrid signal-processing approach combining the TPN and the ANN was proposed to improve the applicability of the ISEs in hydroponics by effectively managing the signal drift and the interference. The parameters of the method were optimized by the 27 training samples, which imitated the hydroponic background. The feasibility and the performance of the method was validated through eight of the real hydroponic sample applications.

From the results, the conventional processing methods such as the TPN and the ANN were sometimes unsatisfactory for prediction of the ion concentrations in hydroponic samples due to their vulnerability to the interference or the drift. The hybrid method improved the RMSEs to 47.2, 13.2, 18.9, and 14.6 mg∙L^−1^, which were approximately half the values of the conventional methods, with CVs below 10% for NO_3_, K, Ca, and Mg, respectively. Furthermore, the hybrid method showed potential as an approximate diagnostic tool for Mg prediction despite the lack of direct Mg ISEs in the sensor array. The structure of the hybrid method can be utilized fundamentally for other ISEs. Therefore, the TPN-ANN method enables the ISEs to measure the individual ions in hydroponic solutions by minimizing the effects of signal drifts and the interference, thereby contributing to both improved efficiency of fertilizer use and reduced environmental pollution when growing plants in closed hydroponics.

## Figures and Tables

**Figure 1 sensors-19-05508-f001:**
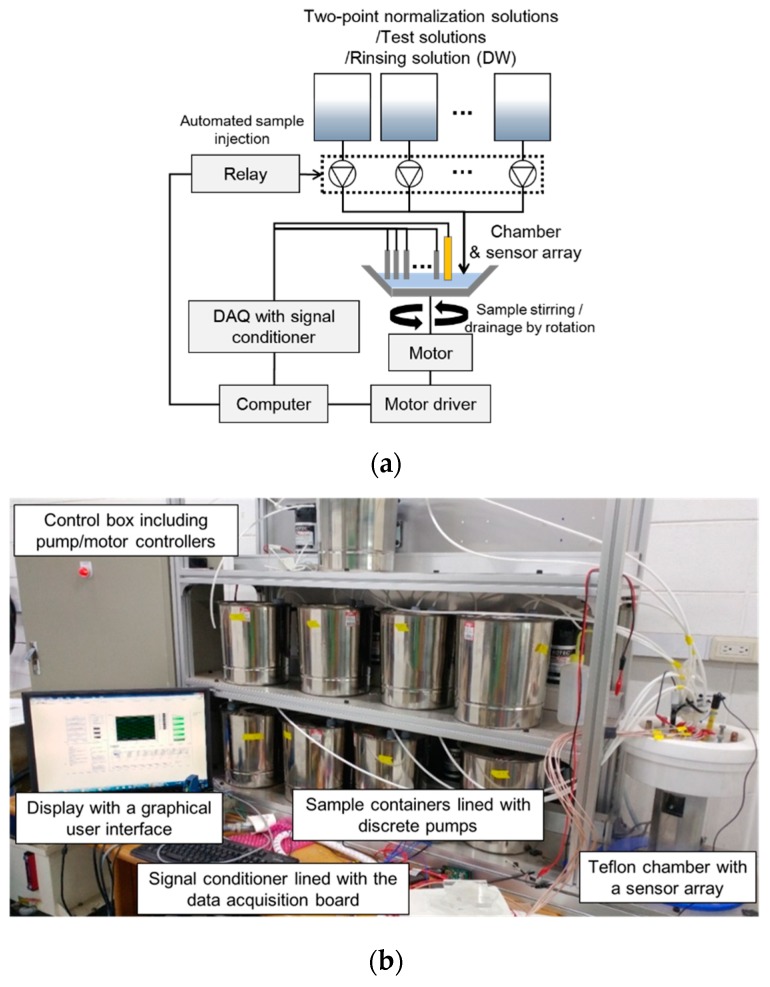
View of the schematic diagram of the test stand (**a**) and the automated test stand (**b**).

**Figure 2 sensors-19-05508-f002:**
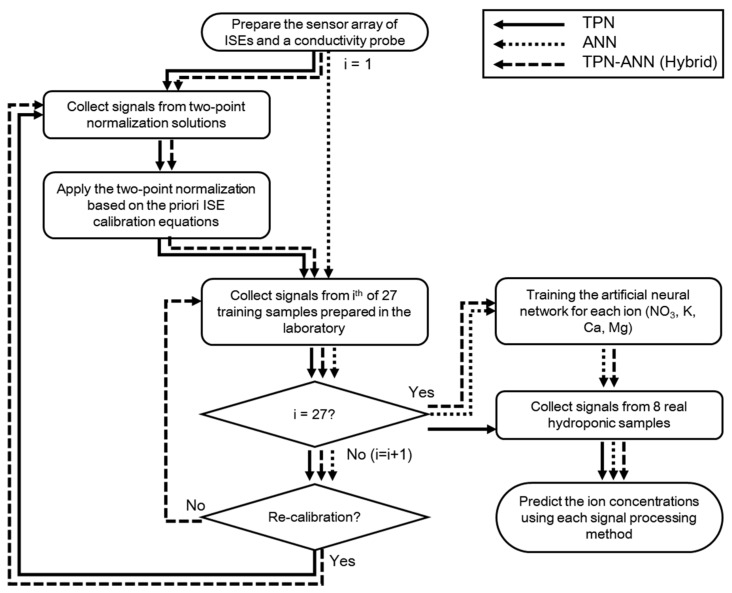
Block diagram of the sample measurement process.

**Figure 3 sensors-19-05508-f003:**
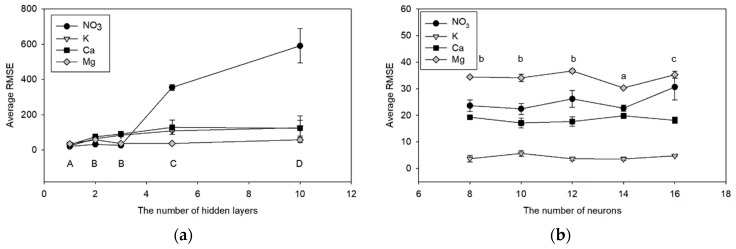
Trends of the root mean square errors (RMSEs) according to the number of hidden layers (**a**) and hidden neurons (**b**). Error bars indicate the standard deviations of three replicates (n = 3, Duncan’s multiple range test, a~c: *p* < 0.05, A~D: *p* < 0.01).

**Figure 4 sensors-19-05508-f004:**
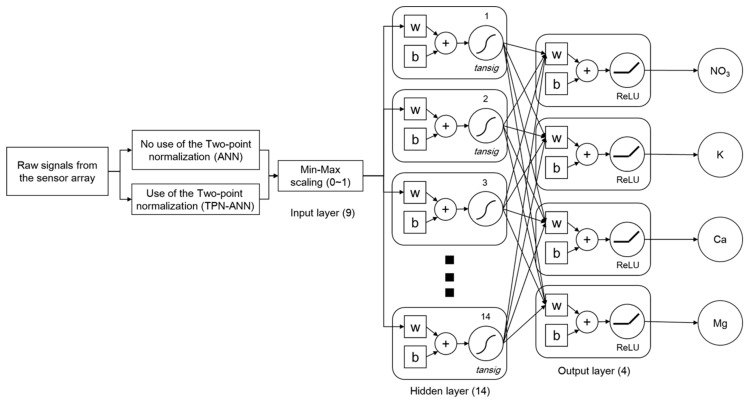
Diagram of the determined neural network structure for artificial neural network (ANN)-based methods (w: weight value, b: bias).

**Figure 5 sensors-19-05508-f005:**
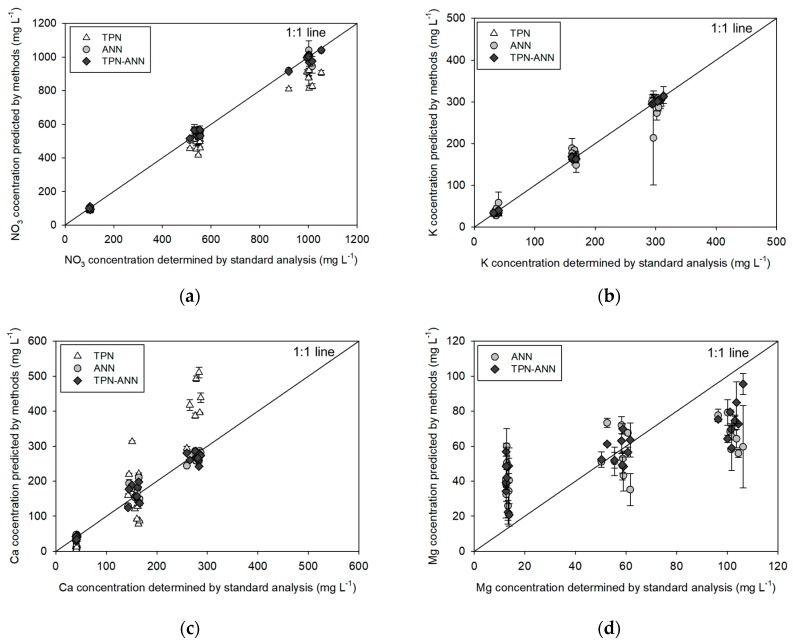
Relationships between ion concentrations determined by the sensor array with three data processing methods and standard analyzers: (**a**) NO_3_, (**b**) K, (**c**) Ca, and (**d**) Mg. Error bars indicate standard deviations of three replicates.

**Figure 6 sensors-19-05508-f006:**
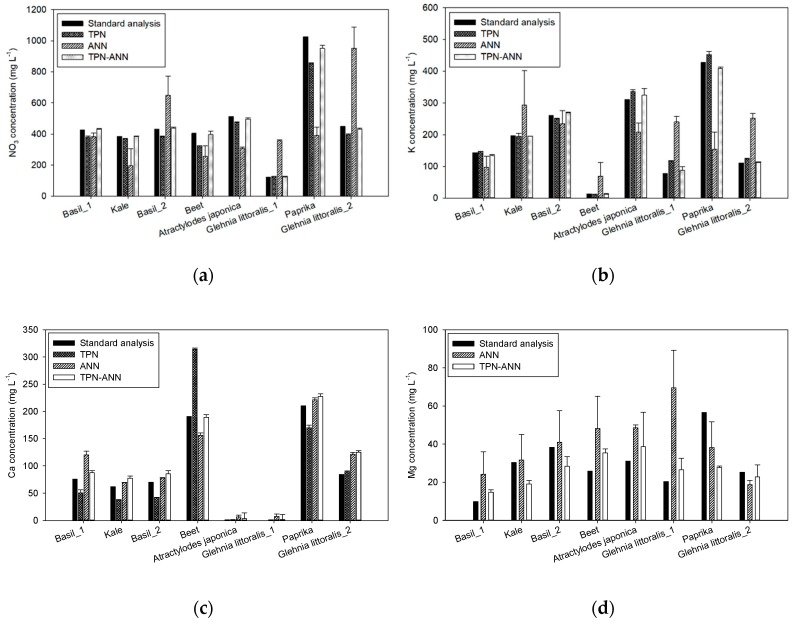
Comparisons of the actual concentrations with the predicted concentrations by three signal-processing methods using 8 different hydroponic samples: (**a**) NO_3_, (**b**) K, (**c**) Ca, and (**d**) Mg. Error bars indicate standard deviations of three replicates.

**Figure 7 sensors-19-05508-f007:**
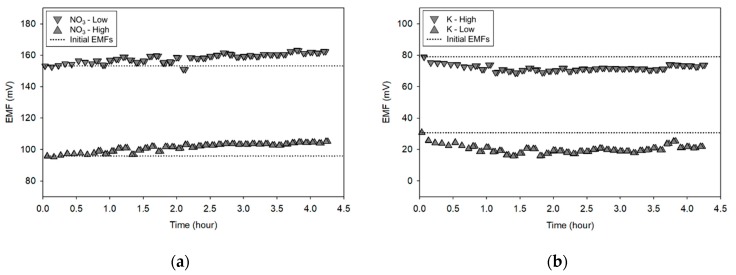
Representative electromotive force (EMF) values showing drifts of (**a**) NO_3_, (**b**) K, and (**c**) Ca ISEs from two-point normalization during the measurement (‘Low’ and ‘High’ in legends indicate the EMF values from the low and high concentrations of two-point normalization solutions, respectively).

**Table 1 sensors-19-05508-t001:** Chemical compositions of NO_3_ and K ion-selective electrode (ISE) membranes used in this study ^[a]^.

Component	NO_3_		K	
	Reagent	Composition	Reagent	Composition
Ionophore	TDDA	4.0% (8 mg)	Valinomycin	2.0% (4 mg)
Plasticizer	NPOE	67.75% (135.5 mg)	Dos	64.7% (129.4 mg)
Matrix	PVC	28.25% (56.5 mg)	PVC	32.8% (65.6 mg)
Ionic additive			KTClPhB	0.5% (1 mg)

^[a]^ TDDA = tetradodecylammonium nitrate, DOS = bis(2-ethyhexyl) sebacate, NPOE = 2-nitrophenyl octylether, PVC = high-molecular-weight polyvinyl chloride, and KTClPhB = potassium tetrakis(4-chlorophenyl)borate.

**Table 2 sensors-19-05508-t002:** Performance characteristics of the NO_3_, K, and Ca ISEs reported in the previous studies.

Sensor	Linear Range (mg∙L^−1^)	Detection Limit (mg∙L^−1^)	Response Time (s)	Lifetime (days)	References
NO_3_	3–1600	3	~50	~60	[17,23,26,27]
K	3–700	3	~50	~60	[17,23,26]
Ca	3–700	3	~50	~40	[17,23]

**Table 3 sensors-19-05508-t003:** Hydroponic samples used in this study.

Sample	Growing Period	Hydroponic System	Nutrient Solution Recipe	Sampling Sites
Basil 1	5 weeks	Deep Flow Technique (DFT) (closed)	Yamazaki’s hydroponic nutrient solution	Experimental farm of Seoul National University (SNU)
Kale	3 weeks	Nutrient Film Technique (NFT) (closed)	Otsuka House’s hydroponic nutrient solution	Smart farm of Korea Institute of Science and Technology (KIST)
Basil 2	5 weeks	DFT (closed)	Yamazaki’s hydroponic nutrient solution	Experimental farm of SNU
Beet	5 weeks	NFT (closed)	Otsuka House’s hydroponic nutrient solution	Smart farm of KIST
*Atractylodes japonica*	6 weeks	NFT (closed)	Hoagland’s hydroponic nutrient solution	Plant factory of Jeju National University (JNU)
*Glehnia littoralis* 1	8 weeks	NFT (closed)	Hoagland’s hydroponic nutrient solution	Plant factory of JNU
Paprika	14 weeks	Drip Irrigation (open)	Grodan’s hydroponic nutrient solution	Smart farm of KIST
*Glehnia littoralis* 2	6 weeks	NFT (closed)	Hoagland’s hydroponic nutrient solution	Plant factory of Chungbuk National University

**Table 4 sensors-19-05508-t004:** Correlation between the predicted concentrations with the actual concentrations for NO_3_, K, Ca, and Mg.

Ion	Processing Method	Linear Relationship ^[a]^	Confidence Intervals for Regression Slope		Coefficient of Determination (R^2^)	RMSE ^[b]^ (mg∙L^−1^)
			Lower 95%	Upper 95%		
NO_3_	TPN	Y = 0.87X + 6.07	0.846	0.889	0.99	89.1
	ANN	Y = 1.00X + 0.52	0.984	1.014	0.99	22.3
	TPN-ANN	Y = 1.00X + 2.7	0.981	1.006	0.99	19.2
K	TPN	Y = 1.01X + 0.28	0.988	1.025	0.99	9.3
	ANN	Y = 0.94X + 7.18	0.884	1.005	0.97	26.3
	TPN-ANN	Y = 1.00X − 0.12	0.992	1.007	0.99	3.7
Ca	TPN	Y = 1.57X − 62.46	1.364	1.768	0.82	93.0
	ANN	Y = 0.97X + 6.4	0.918	1.02	0.97	18.0
	TPN-ANN	Y = 0.96X + 4.41	0.908	1.014	0.97	18.9
Mg	ANN	Y = 0.29X + 37.47	0.186	0.392	0.51	29.3
	TPN-ANN	Y = 0.4X + 34.79	0.306	0.485	0.69	24.9

^[a]^ X represents the concentrations predicted by the processing methods and Y represents the concentrations determined by the standard analysis. ^[b]^ RMSE = $\sqrt{\frac{\sum_{i}^{N}{\left(\hat{x_{i}}-x_{i}\right)}^{2}}{N}}$; where $\hat{x_{i}}$: concentration estimated by ISE, $x_{i}$: actual concentration determined by standard instruments, N: number of sample measurements. TPN: two-point normalization.

**Table 5 sensors-19-05508-t005:** Comparison of processing methods to predict NO_3_, K, Ca, and Mg concentrations in hydroponic samples.

Predicted Ion	Conc. Range (mg∙L^−1^)	Processing Method	Accuracy (RMSE, mg∙L^−1^)	Precision (CV ^[a]^, %)
NO_3_	120–1025	TPN	75.4	1.1
		ANN	133.5	17.9
		TPN-ANN	47.2	2.9
K	13–430	TPN	19.8	2.4
		ANN	144.7	30.1
		TPN-ANN	13.2	4.6
Ca	0–210	TPN	48.8	3.3
		ANN	26.1	13.8
		TPN-ANN	18.9	6.6
Mg	10–60	TPN	Not measurable	
		ANN	29.4	26.6
		TPN-ANN	14.6	28.6

^[a]^$\mathrm{CV}=\frac{SD}{\bar{x}}\times 100$; $\mathrm{SD}=\;\sqrt{\frac{\sum_{i}^{N}{\left(\hat{x_{i}}-{\bar{x}}_{sample}\right)}^{2}}{N-1}}$; where $\hat{x_{i}}$: concentration estimated by ISE, ${\bar{x}}_{sample}$: average concentration estimated by ISE for each sample, N: number of sample measurements, $\bar{x}$: average concentration of N measurements.

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
