# Peer review of "Hybrid Signal-Processing Method Based on Neural Network for Prediction of NO3, K, Ca, and Mg Ions in Hydroponic Solutions Using an Array of Ion-Selective Electrodes"

_sensors, 2019, doi:10.3390/s19245508_

Round 1

Reviewer 1 Report

Nutrient monitoring is a challenging scientific issue in Hydroponics. The authors conducted continuous researches in this area and reported plenty of great progressed on ISEs based nitrate-nitrogen and mineral potassium measurement in soil and hydroponics.

After reviewing, I have plenty question concerning about the paper

The ISE performance

According to the results reported, “From the application test using 8 samples from real greenhouses, the hybrid method 31 showed the root mean square errors of 47.2, 13.2, 18.9, and 14.6 mg∙L-1 over the range of 120-1025, 32 13-430, 0-210, and 10-60 mg∙L-1 for NO3, K, Ca, and Mg, respectively, which were the best prediction 33 results when compared to the conventional approaches.”

The detection precision seemed to be not acceptable for the horticulture application. Did the ISEs work normally? How about their electrochemical performance, e.g. detection range, limit, response time, repeatability and selectivity? Whether the sensor, the author used, was suitable for hydroponic detection, or not?

The author should give enough information in the “material and method” part.

About TPN and ANN

The author had been conducted plenty of researches on both of these two topics according to the Reference. What is the innovation in this paper? The author had to address it clearly.

What kind of ANN did the authors choose? Why don’t you choose other types of machine learning methods?

The Hydroponic experiments

The authors had conducted plenty of work on hydroponic nutrients monitoring. According to the paper, monitoring experiments on the 8 hydroponic plants, or 8 experimental cycles of different plants, had been conducted. However, less information about the cultivation design was introduced. What’s the n ingredient of the nutrient solution? What’s the difference between Basil-1 and Basil-2? Did the ISEs produce a similar performance among those tests? What’s the sample volume in each of these tests? How did the sample collect? Whether the sample was representative?

Other questions

There is a lack of explanation of replicates and statistical methods used in the study.

The discussion part was confusing. The result was presented in the "EMF vs Testing Duration" method. It was hard to tell any useful information with the raw data.

Author Response

Response to Reviewer 1 Comments

We appreciate the reviewer for careful comments and suggestion to improve the quality of the manuscript. According to the comments, the manuscript has been revised.

Here is a listing of the reviewer comments along with our responses to the comments. All line numbers in the responses are based on the finalized form of the revised manuscript (Simple Markup in Microsoft Word).

Point 1 (The ISE performance): According to the results reported, “From the application test using 8 samples from real greenhouses, the hybrid method showed the root mean square errors of 47.2, 13.2, 18.9, and 14.6 mg∙L-1 over the range of 120-1025, 13-430, 0-210, and 10-60 mg∙L-1 for NO3, K, Ca, and Mg, respectively, which were the best prediction results when compared to the conventional approaches.”

The detection precision seemed to be not acceptable for the horticulture application. Did the ISEs work normally? How about their electrochemical performance, e.g. detection range, limit, response time, repeatability and selectivity? Whether the sensor, the author used, was suitable for hydroponic detection, or not?

The author should give enough information in the “material and method” part.

Response 1: We have revised the abstract (lines 27-33) and the discussion (lines 362-368) to emphasize the main findings of our research more clearly. Specifically, we think that the measurable concentration ranges (Table 5), the coefficients of variance (CVs, Table 5), and the EMFs of the ISEs (lines 307-315, Figure 7) would show useful information about how good the hybrid method would be for use in hydroponic applications in terms of detection limit and precision. Following the comments suggested by the reviewer, information about the measurement range, detection limit, response time, and lifetime of the ISEs used in the study has been added in the ‘Materials and Methods’ section by referring to the results of previous studies (lines 121-123, Table 2).

Point 2 (About TPN and ANN): The author had been conducted plenty of researches on both of these two topics according to the Reference. What is the innovation in this paper? The author had to address it clearly.

Response 2: Thanks for the good comments. We agree with the reviewer’s opinion about the innovation of the study should be addressed more clearly. The innovation of our study was the development of a sensor signal processing method that not only can effectively compensate for signal drifts and interferences from other ions, but also can predict Mg ions without a commercially available ionophore for selective recognition of Mg, which were major limitations in the use of the conventional TPN and ANN methods. We have revised the sentences in lines 77-84 and 91-103 to deliver this point more clearly.

Point 3 (About TPN and ANN): What kind of ANN did the authors choose? Why don’t you choose other types of machine learning methods?

Response 3: As requested, we have revised the sentences to describe the use of a feed-forward backpropagation neural network (lines 144-145) and a better flexibility of ANN in solving problems by controlling the number of hidden neurons and hidden layers over other machine learning methods (lines 89-90).

Point 4 (The Hydroponic experiments): The authors had conducted plenty of work on hydroponic nutrients monitoring. According to the paper, monitoring experiments on the 8 hydroponic plants, or 8 experimental cycles of different plants, had been conducted. However, less information about the cultivation design was introduced. What’s the n ingredient of the nutrient solution? What’s the difference between Basil-1 and Basil-2? Did the ISEs produce a similar performance among those tests? What’s the sample volume in each of these tests? How did the sample collect? Whether the sample was representative?

Response 4: Since we came to realize that information about the hydroponic samples used in the study was insufficient, we have added sentences to give more detailed information about the hydroponic samples including the cultivation system, the meaning of the n ingredient of the nutrient solutions, and the sample collection method (lines 196-200, Table 3). For the information about the ISEs during the tests, we have revised a figure for the sensor signals during the measurements (Figure 7) and descriptions (lines 307-315) for sensitivities. Table 5 was also revised to add information about the error statistics obtained with the three different methods. Finally, the sample volume used for each of the tests has been mentioned in line 216.

Point 5 (The Hydroponic experiments): There is a lack of explanation of replicates and statistical methods used in the study. The discussion part was confusing. The result was presented in the "EMF vs Testing Duration" method. It was hard to tell any useful information with the raw data.

Response 5: We have included information regarding the replicates and statistical analysis such as RMSE and CV used in the study in lines 223-225, 282-283, and 318-320. In addition, we have revised the ‘Discussion’ part and the graph of "EMF vs Testing Duration" has been moved to ‘Results’ part and revised to deliver information about the sensor status during the experiments (lines 307-315, Figure 7).

Reviewer 2 Report

1. Could the authors correct a name of ion-selective membranes presented in table 1? 2. The training samples for the sensing array contain chloride ions, which influences nitrate sensing. Could the authors comment on the interference of chloride on nitrate sensor? 3. Could the authors explain why in some solutions shown in Fig. 6b, predicted concentration for potassium was far higher than a value measured by the standard analysis? 4. Please add unit of the ion concentrations to a table A1 in Appendix A.

Author Response

Response to Reviewer 2 Comments

We appreciate the reviewer for careful comments and suggestion to improve the quality of the manuscript. According to the comments, the manuscript has been revised.

Here is a listing of the reviewer comments along with our responses to the comments. All line numbers in the responses are based on the finalized form of the revised manuscript (Simple Markup in Microsoft Word).

Point 1: Could the authors correct a name of ion-selective membranes presented in table 1?

Response 1: Thanks for the careful observation. The names of ion-selective electrodes have been revised in Table 1.

Point 2: The training samples for the sensing array contain chloride ions, which influences nitrate sensing. Could the authors comment on the interference of chloride on nitrate sensor?

Response 2: That’s a good point. We have added sentences to discuss the effect of the interfering ions including Cl on NO3 measurements in lines 334-338.

Point 3: Could the authors explain why in some solutions shown in Fig. 6b, predicted concentration for potassium was far higher than a value measured by the standard analysis?

Response 3: Thanks for the good comments. We have revised the sentences to describe the prediction errors of the ANN methods in lines 307-315.

Point 4:  Please add unit of the ion concentrations to a table A1 in Appendix A.

Response 4: As suggested, we have added the unit of the ion concentrations to the caption of Table A1. Thank you.

Reviewer 3 Report

This is a very interesting paper, which recognises that more advanced analysis methods are needed for broader applications of ISEs. The proposed TPN-ANN method is a compelling approach to address some of the main challenges in real-life settings: drift and interference.

Although artificial neural networks are not used by many ISE researchers at present, the authors do a very good job explaining the approach and have included diagrams (Figure 2 and 4) to help those less familiar with ANNs.

However, there are some changes to the text that would improve the paper and make it more accessible to ISE researchers.

Line 44: “…the nutrient solutions flow into the growing bed once and the remaining solutions are discarded…”. Please clarify. Do you mean that the nutrient solutions “flows through the growing bed and is discarded…” Line 73: The main aim of the paper is to demonstrate the benefits of the TPN-ANN method, but the description of TPN is limited to a single sentence. Please expand to make the approach more clear and better define methods, such as “sensitivity adjustment”. 1-2 sentences should be sufficient. Line 79: Include the simplified Nernst equation, which would then allow you to link back to this equation in line 124 (i.e. with a parameter instead of just “slope” or potential”). Line 94: “application” should be “applications”. Line 98: “were” should be “are”. Section 2.1: Include the LOD for each ISE. Table 1: “KTpClPhB” appears to be a typo. This compound is usually referred to as KTClPhB. Line 132: what do you mean by “…a multiple hidden layer…”? Do you mean “multiple hidden layers”? Re-write this sentence for clarity. Line 133: “could be increased” is a vague phrase. You are talking about training time - could you say that it “is increased”? Or “is usually increased”? Line 137: “were set to 1” should be “was set to 1” Line 139: “…to select the optimal number of the hidden layer”. This could have two meanings. Do you mean “…to select the optimal number of hidden layers.”? Please clarify. Line 142-143: “…were used considering the variation in training” is unclear and unnecessary. Instead, I suggest something like “ …was evaluated based on RMSEs of three replicate training results”. Line 148: briefly define “activation function”. Line 174: replace “Hoagland’s solution” with “Hoagland’s hydroponic nutrient solution”. Line 181: it would be more clear if “would include a wide range” was replaced with “spans a wide range”. Line 188: replace “modified based on” with “modified from” or “based on” or something similar. Line 191: it is unclear what is meant by “computer system LINED with a signal conditioning…”. “lined” could likely be deleted. Move Table 2 to the appendix. Line 213: I was surprised by the results in Figure 3a (i.e. RMSE increases substantially with the number of hidden layers). Please include a sentence or two discussing this, or giving a reference that confirms it is not unexpected. Section 3.2. Throughout this section, the small differences are given undue weight, especially given the amount of error. For example, (a) the difference between slopes is a major part of the comparison. However, slopes are given without uncertainty, even though the data has significant noise. The 95% CI for each slope should be calculated in presented in Table 3. (b) RMSE is presented with an inappropriately high number of significant figures, which makes the measurements appear misleadingly precise. Although more complex methods, such as bootstrapping, could be used to estimate RMSE variability (and thus an appropriate number of significant figures), a simpler approach could be used based on accuracy. For example, estimate the appropriate number of significant figures for a sample, then using that same number of significant figures for the RMSE. Specifically, you don’t know the sample concentration to one decimal place, so it is misleading to report it so precisely. Section 3.3: the second sentence “In comparisons…” is hard to follow. Please re-write for clarity. Table 4: See comment 20 above. Lines 293 – 301. Although this is the Discussion section, this paragraph is a summary of the results and not appropriate (or necessary) for this section. Line 303: What is meant by “allowable deviations and errors”? Lines 315 – 316: “TPN-ANN > TPN > ANN” is unusual notation, given that you are not specifically referring to numeric values. Instead, I suggest something like “…the processing methods showed that TPN-ANN was the best predictor, followed by TPN, then ANN. Line 317: I suggested “TPN is vulnerable to interferences”. Line 317: Given my suggestions for rewording this statement (above), I believe the data is over interpreted. Specifically, the authors believe that the results in Figure 6(c) demonstrate TPN’s susceptibility to interference. However, the higher RMSE appears to be driven by poor performance for the Beet samples. While for samples with a less high concentration, TPN outperforms ANN, e.g. Basil 1, Atractylodes, Glehnia. However, these low-concentration samples are down weighted. Lines 320-323 and Figure 6: this should be in the Results section, not discussion. Line 327: I suggest “various background ions” Finally, although the Conclusions captures the importance of the work, with its over-reliance on RMSE’s and concentrations the abstract does not capture this excitement. As a result, I suggest re-writing the last few sentences to highlight the overarching message, instead of simply a list of numeric results and experimental details.

There are also some small changes that should be made to the figures to make them easier to interpret. In particular, many of the figures do not print out well in black and white, even though that is how many researchers read their papers.

First, I’d like to point out that the diagrams in Figure 2 and Figure 4 are especially well done. Figure 3: In the figure NO3 & Ca and K & Mg are almost impossible to tell apart. Colour should be chosen so they are distinguishable when printed in black and white. Indicate the meaning of the error bars (standard deviations?) in the figure description. Figure 6: There is still a problem when printed in black and white. However, given that the bars are always in the same sequence, this is OK. The figure should be adjusted to fit on a single page. Figure 7: Ensure it fits on the same page The x-axis is in a fairly useless unit (0 – 16,000 seconds). Changing this to hours would be more meaningful. I also suggest you add the word “drift” to the figure caption, to help people who are just quickly skimming the article.

Finally, I would like to reiterate that this is a very good paper that presents a method that should be considered by more researchers. And, although English does not appear to be the primary language of the authors, the manuscript is generally very clear and easy to follow. By making the above changes, I believe the paper will reach a broader audience.

Author Response

Response to Reviewer 3 Comments

We appreciate the reviewer for careful comments and suggestion to improve the quality of the manuscript. According to the comments, the manuscript has been revised.

Here is a listing of the reviewer comments along with our responses to the comments. All line numbers in the responses are based on the finalized form of the revised manuscript (Simple Markup in Microsoft Word).

Point 1: There are some changes to the text that would improve the paper and make it more accessible to ISE researchers.

Line 44: “…the nutrient solutions flow into the growing bed once and the remaining solutions are discarded…”. Please clarify. Do you mean that the nutrient solutions “flows through the growing bed and is discarded…”

Response 1: Thanks for the good comment. We agree that “the nutrient solution flows through the growing bed and is discarded” is more clear, so the sentence has been revised in lines 45-46.

Point 2: Line 73: The main aim of the paper is to demonstrate the benefits of the TPN-ANN method, but the description of TPN is limited to a single sentence. Please expand to make the approach more clear and better define methods, such as “sensitivity adjustment”. 1-2 sentences should be sufficient.

Response 2: According to the comments, we have added details of the two-point normalization method in lines 130-135. That is, in principle, in the TPN, individual sensitivity slopes of each of the ISE electrodes are normalized by multiplying the electromotive forces (EMF) data by the ratio of a reference EMF difference to a measured EMF difference using two different solutions with known concentrations of the primary ion corresponding to the electrodes. Offsets are then adjusted by subtraction of the difference between the highest reference point and the modified highest concentration point.

Point 3: Line 79: Include the simplified Nernst equation, which would then allow you to link back to this equation in line 124 (i.e. with a parameter instead of just “slope” or potential”).

Response 3: Following the comment, we have added details of the simplified Nernst equation in lines 135-139 (Equation (1)) and revised “slopes and potentials” by “S, EO, and EJ” in lines 140-141.

Point 4: Line 94: “application” should be “applications”.

Response 4: Thanks for the careful observation. We have replaced “application” by “applications” in line 98.

Point 5: Line 98: “were” should be “are”.

Response 5: Modifications have been made in line 103. Thank you.

Point 6: Section 2.1: Include the LOD for each ISE.

Response 6: According to the reviewer’s comments, we have added a table to give information about the measurement range, detection limit, response time, and lifetime of the ISEs used in the study referring the previous studies (Table 2).

Point 7: Table 1: “KTpClPhB” appears to be a typo. This compound is usually referred to as KTClPhB.

Response 7: Thanks for the careful observation. We have revised the abbreviation of the compound as KTClPhB in Table 1.

Point 8: Line 132: what do you mean by “…a multiple hidden layer…”? Do you mean “multiple hidden layers”? Re-write this sentence for clarity.

Response 8: As requested, we have revised “a multiple hidden layer” by “multiple hidden layers” in line 147.

Point 9: Line 133: “could be increased” is a vague phrase. You are talking about training time - could you say that it “is increased”? Or “is usually increased”?

Response 9: Thanks for the good comment. We have revised “could be increased” by “is usually increased” in line 148.

Point 10: Line 137: “were set to 1” should be “was set to 1”

Response 10: We have revised the sentence following the comment in line 153.

Point 11: Line 139: “…to select the optimal number of the hidden layer”. This could have two meanings. Do you mean “…to select the optimal number of hidden layers.”? Please clarify.

Response 11: According to the reviewer’s comment, we have revised “to select the optimal number of the hidden layer” by “to select the optimal number of hidden layers” in line 155.

Point 12: Line 142-143: “…were used considering the variation in training” is unclear and unnecessary. Instead, I suggest something like “ …was evaluated based on RMSEs of three replicate training results”.

Response 12: According to the reviewer’s comment, we have revised “were used considering the variation in training” by “was evaluated based on RMSEs of three replicate training results” in line 158.

Point 13: Line 148: briefly define “activation function”.

Response 13: To define “activation function” more clearly, we have added a short description in lines 165-166.

Point 14: Line 174: replace “Hoagland’s solution” with “Hoagland’s hydroponic nutrient solution”.

Response 14: According to the reviewer’s comment, we have revised “Hoagland’s solution” by “Hoagland’s hydroponic nutrient solution” in line 192.

Point 15: Line 181: it would be more clear if “would include a wide range” was replaced with “spans a wide range”.

Response 15: According to the reviewer’s comment, we have revised “which would include a wide range of ion concentrations” by “which spanned a wide range of ion concentrations” in lines 199-200.

Point 16: Line 188: replace “modified based on” with “modified from” or “based on” or something similar.

Response 16: According to the reviewer’s comment, we have revised “modified based on” by “modified from” in line 210.

Point 17: Line 191: it is unclear what is meant by “computer system LINED with a signal conditioning…”. “lined” could likely be deleted.

Response 17: For more clear expression, we have revised “a main computer system lined with a signal conditioning…” by “a main computer system with a signal conditioning…” in line 213.

Point 18: Move Table 2 to the appendix.

Response 18: According to the reviewer’s comment, we have moved the table to the appendix (Table A2).

Point 19: Line 213: I was surprised by the results in Figure 3a (i.e. RMSE increases substantially with the number of hidden layers). Please include a sentence or two discussing this, or giving a reference that confirms it is not unexpected.

Response 19: That’s a good point. Yu et al. [40] reported that the ANN models with more hidden layers do not guarantee better performance than those with fewer layers if the number of hidden layers is sufficient for the given nonlinear problem. We have added this description in ‘Discussion’ section (lines 328-331).

Point 20: Section 3.2. Throughout this section, the small differences are given undue weight, especially given the amount of error. For example, (a) the difference between slopes is a major part of the comparison. However, slopes are given without uncertainty, even though the data has significant noise. The 95% CI for each slope should be calculated in presented in Table 3. (b) RMSE is presented with an inappropriately high number of significant figures, which makes the measurements appear misleadingly precise. Although more complex methods, such as bootstrapping, could be used to estimate RMSE variability (and thus an appropriate number of significant figures), a simpler approach could be used based on accuracy. For example, estimate the appropriate number of significant figures for a sample, then using that same number of significant figures for the RMSE. Specifically, you don’t know the sample concentration to one decimal place, so it is misleading to report it so precisely.

Response 20: We agree with the opinion that the result should be displayed more concise. We have presented the 95% CI for each slope in Table 3.

Regarding the number of significant figures in sample concentrations, we have used one decimal place since the sample concentrations determined by the standard analysis had values below one decimal place. Regarding the significant figures for samples, we have added information about the low detection limits of standard instruments in lines 201-206.

Point 21: Section 3.3: the second sentence “In comparisons…” is hard to follow. Please re-write for clarity.

Response 21: Following the comment, we have revised the sentence to deliver the meaning more clearly (lines 287-289).

Point 22: Table 4: See comment 20 above.

Response 22: As mentioned above, we have used one decimal place since the sample concentrations determined by the standard analysis were given below one decimal place.

Point 23: Lines 293 – 301. Although this is the Discussion section, this paragraph is a summary of the results and not appropriate (or necessary) for this section.

Response 23: We have revised the paragraph to eliminate the summarized result part and discuss about the effects of the number of hidden layers and neurons on the ANN performance with references (lines 328-333). 

Point 24: Line 303: What is meant by “allowable deviations and errors”?

Response 24: Since we agreed that the “allowable deviations and errors” in the sentence had somewhat vague meaning, the sentence has been revised to be presented based on the linear relationship referring the previous study in lines 334-337.

Point 25: Lines 315 – 316: “TPN-ANN > TPN > ANN” is unusual notation, given that you are not specifically referring to numeric values. Instead, I suggest something like “…the processing methods showed that TPN-ANN was the best predictor, followed by TPN, then ANN.

Response 25: As requested, the text has been revised “TPN-ANN > TPN > ANN” by “the TPN-ANN was the best processing method, followed by TPN and ANN” in lines 350-351.

Point 26: Line 317: I suggested “TPN is vulnerable to interferences”.

Response 26: As requested, we have revised “the Ca prediction by the TPN shows the vulnerability to the interference” by “the Ca prediction by the TPN was vulnerable to interferences” in lines 351-352.

Point 27: Line 317: Given my suggestions for rewording this statement (above), I believe the data is over interpreted. Specifically, the authors believe that the results in Figure 6(c) demonstrate TPN’s susceptibility to interference. However, the higher RMSE appears to be driven by poor performance for the Beet samples. While for samples with a less high concentration, TPN outperforms ANN, e.g. Basil 1, Atractylodes, Glehnia. However, these low-concentration samples are down weighted.

Response 27: Thanks for the good comment. We have added more discussion about the TPN’s susceptibility to interference based on high variations in Ca predictions depending on the samples. This showed that the TPN could be affected by the changes of background ions (lines 352-355).

Point 28: Lines 320-323 and Figure 6: this should be in the Results section, not discussion.

Response 28: Following the comment, we have moved the sentences and Figure 7 to the Results section (lines 307-315, Figure 7).

Point 29: Line 327: I suggest “various background ions”

Response 29: As suggested, we have revised “various background ions” by “various background ions” in line 362.

Point 30: Finally, although the Conclusions captures the importance of the work, with its over-reliance on RMSE’s and concentrations the abstract does not capture this excitement. As a result, I suggest re-writing the last few sentences to highlight the overarching message, instead of simply a list of numeric results and experimental details.

Response 30: We have rewritten the abstract part referring the potential as a Mg prediction and the improvement of the hybrid method for the accuracy and feasibility of ISEs in hydroponic applications to present the highlights of this study more clearly (lines 31-34). Thanks for the good comments.

Point 31: There are also some small changes that should be made to the figures to make them easier to interpret. In particular, many of the figures do not print out well in black and white, even though that is how many researchers read their papers.

First, I’d like to point out that the diagrams in Figure 2 and Figure 4 are especially well done. Figure 3: In the figure NO3 & Ca and K & Mg are almost impossible to tell apart. Colour should be chosen so they are distinguishable when printed in black and white.

Response 31: We agree with the opinion. Figures 3, 5, 6, and 7 have been revised to be distinguishable when printed in black and white.

Point 32: Indicate the meaning of the error bars (standard deviations?) in the figure description.

Response 32: Thanks for the careful observation. We have revised the captions to inform the meaning of the error bars (Figures 3, 5, and 6).

Point 33: Figure 6: There is still a problem when printed in black and white. However, given that the bars are always in the same sequence, this is OK. The figure should be adjusted to fit on a single page.

Response 33: As requested, the figures have been adjusted to be displayed on a single page (Figure 6).

Figure 7: Ensure it fits on the same page. The x-axis is in a fairly useless unit (0 – 16,000 seconds). Changing this to hours would be more meaningful. I also suggest you add the word “drift” to the figure caption, to help people who are just quickly skimming the article.

Response 34: As requested, the unit of the x-axis has been displayed as hours and we have revised the figure caption to include the word “drifts” (line 321, Figure 7).

Round 2

Reviewer 1 Report

The authors had been carefull revised the manuscript. I suggest the acception of this manuscript.